# THE IMPACT OF ENSEMBLE ON HOMOMORPHIC ENCRYPTED DATA CLASSIFICATION

## ABSTRACT

Homomorphic encryption (HE) is encryption that permits users to perform computations on encrypted data without first decrypting it. HE can be used for privacy-preserving outsourced computation and analysis, allowing data to be encrypted and outsourced to commercial cloud environments for processing while encrypted or sensitive data. HE enables new services by removing privacy barriers inhibiting data sharing or increasing the security of existing services. A convolution neural network (CNN) can be homomorphically evaluated using addition and multiplication by replacing the activation function, such as ReLU, with a low polynomial degree. To achieve the same performance as the ReLU activation function, we study the impact of applying the ensemble techniques to solve the accuracy problem. Our experimental results empirically show that the ensemble approach can reduce bias, and variance, increasing accuracy to achieve the same ReLU performance with parallel and sequential techniques. We demonstrate the effectiveness and robustness of our method using three datasets: MNIST, FMNIST, and CIFAR-10 .

## 1 INTRODUCTION

Homomorphic encryption (HE) is a private artificial intelligence (AI) application that allows users to perform computations on encrypted data without decryption, and the result of calculations will be in an encrypted form when decrypted. As a result, HE is ideal for working with sensitive data to privacy-preserving outsourced storage and computation. In other words, HE allows data to be encrypted and outsourced to commercial cloud environments for processing, all while encrypted. Deep learning on the cloud enables designing, developing, and training deep learning applications faster by leveraging distributed networks and HE and cloud computing, allowing large datasets to be easily ingested and managed to train algorithms. It will enable deep learning models to scale efficiently and lower costs. HE scheme adopting bit-wise encryption performs arbitrary operations with an extensive execution time but to shorten execution time, a method adopts a HE scheme to encrypt integers or complex numbers. An HE scheme is usually defined in a finite field, so it only supports two finite field operations, addition, and multiplication, which can behave entirely differently than floating point numbers operations used in typical AI applications. Accordingly, functions commonly used in deep learning, such as ReLU, Sigmod, and max-pooling, are not compatible with HE (Obla, 2020). To address this issue, the polynomial activation function can evaluate CNN to address since HE straightforwardly supports additions and multiplications. Due to the increased complexity in computing circuits with nested multiplications, it is desirable to restrict the computation to low-degree polynomials (Gilad-Bachrach et al., 2016). However, replacing ReLU with a low-degree polynomial, combined with other techniques such as batch normalization (BN) (S.IoffeandC.Szegedy, 2015), still suffers from high bias, variance, and low accuracy. Intuitively, the ensemble is a machine learning approach that consists of a set of individual weak learning models working in sequential or parallel. The outputs are combined with a decision fusion strategy to produce a single and better performance than any single model (Huang et al., 2009). That motivates us to introduce the ensemble approach to enhance accuracy by reducing bias and variance when an HE scheme. Ensemble learning has been recently known to be an essential reason for improving the performance of deep learning models because sub-models do not strongly depend on each other, even though they are trained jointly. Moreover, they exhibit ensemble-like behavior in that their performance smoothly correlates with the number of valid paths and enables them to work with the depth of HE network (Veit et al., 2016). The success of ensembles is due

to the focus on getting a better-performed new model by reducing the bias or variance of weak learners by combining several of them to create a strong learner that achieves better performances. It can be differentiated as bagging will focus on getting an ensemble model with reduced variance, whereas boosting and stacking will produce strong models less biased than their components (Sagi & Rokach, 2018). Generally, ensemble techniques are considered one of the best approaches to better performance due to lower error and overfitting than any individual method, leading to better performance in the test set. If each learner might have a sort of bias, or variance, combining them can reduce this bias. Generally, we can say ensemble is one of the most approaches to better performance (Zhou et al., 2002).

In this paper, we propose an ensemble approach to improve the accuracy of HE-based privacy-preserving data inference with deep neural networks (DNN) for both sequential and parallel ensembles when replacing the ReLU activation function with polynomials. We applied customized sequential ensemble techniques that can be applied to different numbers of CNN models, which will be involved in the multi-class prediction while using polynomials as activation functions in the hidden layer. We applied the bagging method for the parallel ensemble technique and studied the ensemble's impact on bias and variance. Our results indicate that an ensemble could significantly reduce variance and boost accuracy. To the best of our knowledge, this is the first work to investigate the ensemble approach in the context of HE-based privacy-preserving inference with DNN to solve the accuracy problem caused by replacing activation function with polynomials. Most of the previous efforts were focused on choosing a better single polynomial to increase the accuracy. In contrast, our work focuses on improving the low accuracy classification model by combining weak models considering the requirements for encrypted data. Figure 2 illustrates an ensemble approach to increase the accuracy while using polynomials in the hidden layers in the convention network.

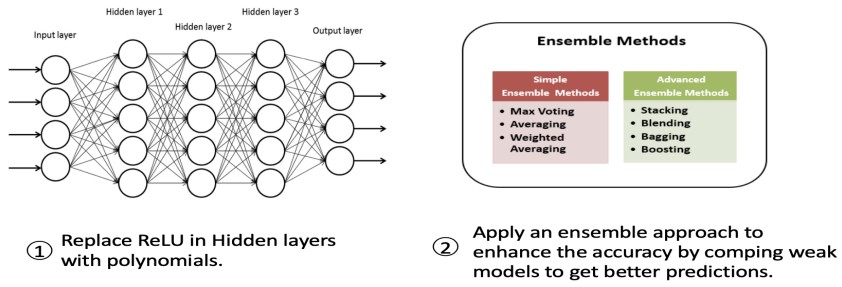

Figure 1: An ensemble approach to increase the accuracy while using polynomials in the hidden layers in the convolution network.

In summary, our contributions to this paper include:

- We have investigated the impact of using the sequential ensemble technique on the accuracy, and the result indicates significant improvement, reaching the same ReLU performance while using polynomials.
- We have studied the impact of the parallel ensemble technique, especially bagging, and our result shows an improvement in the variance result without increasing bias.
- We have demonstrated the effectiveness and robustness of our method using three datasets; MNIST (LeCun, 1998), FMNIST (Xiao et al., 2017), and CIFAR-10 (Krizhevsky et al., 2014).

The rest of the paper is organized as follows: Section 2 reviews related works and summarize the proposed approach's advantages. Section 3 discusses the background. Section 4 presents the details of the proposed method, and Section 5 demonstrates the effectiveness of the proposed approach with experimental results. Finally, we conclusion the paper in Section 6.

## 2 RELATED WORKS

**Replacing the ReLU activation function with polynomial:** To study the impact of replacing the ReLU activation function with the polynomial activation function in neural networks, several works

Table 1: Accuracy performance in related research that have used the same dataset and the type of pooling.

| Reserech | Pooling | BN | Accuracy(%) | | |
|---|---|---|---|---|---|
| | | | MNIST | F_MNIST | CIFAR-10 |
| (Gilad-Bachrach et al., 2016) | Average | - | 98.95 | – | – |
| (Chabanne et al., 2017) | Average | Yes | 99.30 | - | - |
| (Jiang et al., 2018) | Scaled mean | - | 98.10 | - | - |
| (Al Badawi et al., 2018) | Average | - | 99.00 | - | 77.55 |

such as (Jiang et al., 2018) and (Bourse et al., 2018) point out the challenging caused by replacing the ReLU function, and the major challenge is severely decreasing in the accuracy. In 2018, (Al Badawi et al., 2018) tested the inference of a convolutional neural network (CNN) using graphical processing and achieved an accuracy of 99% on the Modified National Institute of Standards and Technology (MNIST) dataset while achieving low classification accuracy (77.55%) (CIFAR)-10 dataset. (Ali et al., 2020) studied a deep network with nine ReLU activation layers. CIFAR-100 datasets, using an improved ReLU and min-max approximation. However, they used a very deep network not capable of HE. Additionally, (Onoufriou et al., 2021) investigated how FHE with deep learning can be used at scale toward accurate sequence prediction. They reached an accuracy of 87.6% on average; however, they used confidential data that cannot be made public. Some work such as (Gilad-Bachrach et al., 2016), (Jiang et al., 2018), and (Chabanne et al., 2017) only studied the accessible gray-scale dataset MNIST because it is easy to classify with high accuracy. Table 1 illustrates the accuracy of each research in the same dataset that we have used.

**Ensemble techniques:** Numerous works studied the impact of the ensemble approach in solving the problem of bias and variance. They point out that one of the main reasons behind the success of the ensemble methods is combining multiple ways, such as averaging and voting, where the ensemble model performs better than any of the individual models. (Dietterich, 2000) reviewed ensemble methods and explained why ensembles could often perform better than any single model. The reasons for the success of ensemble learning include: statistical, computational, and representation learning. Bias-variance decomposition of the expected misclassify rate, the most commonly used loss function in supervised learning and frequency-based estimation of the decomposition terms, is biased and shows how to correct for this bias and study the decomposition of various algorithms and datasets (Kohavi et al., 1996). (Webb & Zheng, 2004), investigated the different mechanism that allows for trade-offs to the bias-variance decomposition of the error ensemble methods, such as the random forest and application of post-processing algorithms. (Breiman, 2001) showed heuristically that the variance of the bagged predictor is less than the original predictor and proposed that bagging is better in higher dimensional data. (Bühlmann & Yu, 2002) explained how bagging gives smooth, slight variance and mean bias error, introducing sub bagging and half sub bagging. A high-bias model is caused by a low degree of the polynomial will result in underfitting. In contrast, high variance is caused when the model has many degrees of freedom, like a high-degree polynomial model, and causes overfitting.

## 3 BACKGROUND

Machine learning aims to build a model that performs well with the training and the testing data.It is crucial to understand prediction errors, bias, and variance, then a trade-off between a model's ability to reduce bias and variance (Geman et al., 1992), (Brown et al., 2005), (Domingos, 2000), (Rocks & Mehta, 2022). The proper calculation of these errors would increase the accuracy and avoid the mistake of overfitting and underfitting, both leading to poor predictions (Bullinaria, 2015), (Dong et al., 2020).

- **Overfitting:** A model performs well on the training but does not perform well on the test set; overfitting occurs if the model shows low bias but high variance. Training the model with so much data will learn from our dataset's noise and inaccurate data entries. As a result, the model does not categorize the data correctly.

- **Underfitting:** It occurs when the machine learning algorithm cannot capture the data trend. Intuitively, underfitting happens when the model or algorithm does not sufficiently fit the data. Specifically, underfitting occurs if the model or algorithm shows low variance but high bias.

The error consists of the essential three components: Bias and variance, and irreducible errors are as follow:

$$Error(x) = Bais^2 + Variance + IrreducibleError$$

To build a good model, we need to find a good balance between bias and variance such that it optimizes the total error. Since we can't control Irreducible error, we can handle bias and variance by minimizing the sum of the bias and variance contributions to the generalization error. The definition of Bias Error and Variance Error is as follows:

- **Bias Error:** Bias is the average difference between predicted and actual results. High bias means we are getting low-performance Under-fitting.
- **Variance Error:** Quantify the expected value difference in the same way observation on that time model is over-fitting. It is caused by a very complex model on simple data. Train a model that shows high variance near 100% accuracy on training data. However, checking the present data model fails to predict the correct result.

Managing bias and variance is crucial to achieving a well-performing training model prediction. There will be a trade-off between minimizing the bias and minimizing the variance as given (Geman et al., 1992), (Brown et al., 2005), (Domingos, 2000) :

$$E[o - t]^2 = bias^2 + \frac{1}{M}var + (1 - \frac{1}{M})covar$$

$$bias = \frac{1}{M}var \sum_i E(o_i - t)$$

$$var = \frac{1}{M}var \sum_i E[o_i - E[O_I]]^2$$

$$covar = \frac{1}{M(M-1)} \sum_i \sum_{j \neq i} E[o_i - E[O_i]][o_j - E[O_j]]$$

Where $t$ is each model's target $o_i$ output, and $M$ is the ensemble size. Here, the bias term measures the average difference between the base learner and the model output, var indicates their average variance, and covar is the covariance term measuring the pairwise difference of different bases. According to (Bylander & Tate, 2006) and (Loughrey & Cunningham, 2005), there is no single solution to avoid overfitting. However, we can use techniques to eliminate overfitting, such as increasing the data size, cross-validation, early stopping, regularization, and ensemble technique.

## 3.1 ENSEMBLE TECHNIQUE

Neural network models are a nonlinear method that enables it to learn complex nonlinear relationships in the data. However, it will make the models extremely sensitive to initial conditions in terms of random weights and the statistical noise in the training dataset. Accordingly, every time the model is trained, it will learn a different version from the mapping input and output. As a result, the ensemble can successfully reduce the variance of neural network models by combining the perdition of multiple models instead of a single model to ensure that the most stable and best possible prediction is considered. According to (Krizhevsky et al., 2017), the performance of one model compared to ensemble predictions averaged over two, five, and seven models are distinct. Averaging the predictions of five similar CNN's gives an error rate of 16.4%, whereas averaging the predictions of two pre-trained CNNs with the same five CNN's gives an error rate of 15.3%. Indeed, an ensemble improves the quality of essential algorithms and increases the underlying algorithms' diversity. Also, it aims to make it sufficiently diverse, and the correct operation of other algorithms will compensate for the errors of individual algorithms. The essential idea of ensemble methods is to reduce the bias-variance of weak learners by combining several of them to create a strong learner called the ensemble model that achieves better performances. The vulnerable learners or base models can be used as building blocks for designing more complex models by combining several. The primary models' performance is feeble because they are of high bias due to a low degree of freedom models and high variance to be robustly caused by the high degree of freedom models (Rocca, 2019). The number of models in the ensemble is essential to keep small due to computational expense in training models and the diminishing returns in performance from adding more ensemble members. We can use different approaches, such as various datasets, models, or combinations.

## 3.2 CATEGORIES OF ENSEMBLE METHODS

The ensemble strategies are broadly categorized into sequential ensemble and parallel ensemble techniques:

- **Sequential ensemble techniques:** Generate base learners in a sequence, creating a dependency between each base learner. The model's performance is improved by assigning higher weights to previously misrepresented learners (Odegua, 2019).
- **Parallel ensemble technique:** The base learners are created in a parallel format where the methods take advantage of the similar generation of base learners to encourage independence, significantly reducing the error between the base learners. Most ensemble techniques apply a single algorithm in base learning, resulting in homogeneity in all base learners. Other methods involve heterogeneous base learners, giving rise to heterogeneous ensembles, which means base learners are learners of distinct types. In contrast, homogeneous base learners are base learners of the same kind with similar qualities.

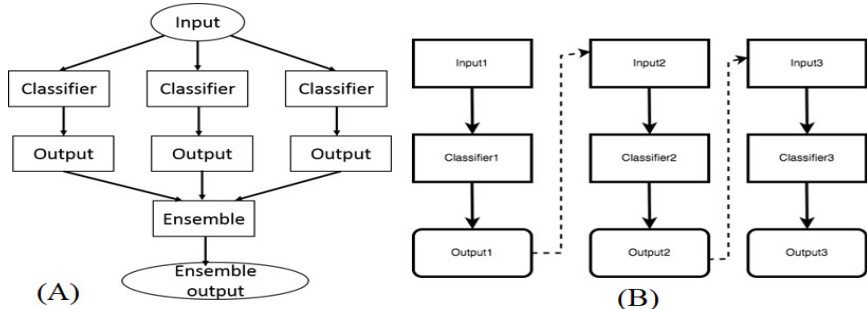

Figure 2: An ensemble approach to increase the accuracy while using polynomials in the hidden layers in the convolution network (A)Sequential ensemble technique (B) Parallel ensemble technique

Generally speaking, existing ensemble learning methods can be grouped into three types:

- **Bagging**: Homogeneous weak learners independently learn from each other in parallel and combine them following some deterministic averaging process (Breiman, 1996). The model is called stable when it is less sensitive to small fluctuations in the training date (Alelyani, 2021). Bagging is more useful when the model is unstable, while bagging is not valuable for improving the performance in stable models.
- **Boosting**: Homogeneous weak learners sequentially learn in an adaptive way where the base model depends on the previous model and combines them following a deterministic strategy (Freund, 1995).
- **Stacking**: Heterogeneous weak learners, learning in parallel and combining and training a meta-model and the output prediction based on the different weak models' predictions (Wolpert, 1992).

But other variant methods are also possible based on specific problem needs.

## 3.3 ADVANTAGE AND DISADVANTAGES OF ENSEMBLE LEARNING

Advantages of Model Ensembles :

- Ensemble methods have higher predictive accuracy compared to the individual models.
- Ensemble methods are beneficial when there is both linear and non-linear type in the dataset; by combining different models to handle this type of data.
- With ensemble methods, bias and variance, can be reduced, which avoids underfitting and overfitting issues and makes it less noisy with more stability.

Ensembles are not always better because they cannot help unknown differences between the sample and the population. Ensembles should be used carefully due to their difficulty in interpreting.

Using the ensemble approach should consider the cost of creating and computing, increasing the complexity of the classification, deploying the ensemble carefully, and being hard to implement in a real-time platform. Generally, ensembles provide higher predictive accuracy by creating lower variance and bias and providing a deeper understanding of the data. However, selecting the model and the suitable ensemble methods is crucial since wrongly choosing the technique will lead to lower predictive accuracy than an individual model.

## 4 PROPOSED METHOD

Our proposed method is to replace the ReLU activation Function that is not capable of HE in the hidden layers with a polynomial function, then apply the ensemble technique as an extra step to boost the accuracy of the classification model. We believe the ensemble will solve the accuracy problem caused by using the polynomial activation function in hidden layers. We study the accuracy for both parallel and sequential.

### 4.1 SEQUENTIAL ENSEMBLE TECHNIQUES

We create an ensemble of multiple CNN models. This ensemble is applied to the multi-class prediction of the three datasets. First, we define the individual CNN model, which is trained sequentially. Every individual model gives its prediction, then the final prediction of the ensemble model will be the most frequent prediction by all the individual CNN models. Only ReLU functions in the hidden layer are replaced:

$$\sigma_r(x) = max(x, 0)$$

With the low-degree polynomial activation function:

$$\sigma_{poly}(x) = x^2 + x$$

We choose degree-2 polynomials according to (Ali et al., 2020) it will lead to better performance, and a higher than degree-2 polynomials would result in overfitting.

**Architecture of convolution network:** We train on MNIST, FMNIST, and CIFAR-10 with polynomial activation functions. The structure of the CNN model is as follows:

- Two convolutional layers: first layer with 32 filters of size 3 x 3, second 46 filters of size 3 x 3.
- Apply after every convolutional layer average pooling and batch normalization.
- After a dropout layer, the network results are parsed from a final fully connected layer of ten neurons.

### 4.2 PARALLEL ENSEMBLE TECHNIQUES

We use a parallel ensemble approach bagging to study the impact of variance and bias due to the ease of implementation using Python libraries such as scikit-learn and the ease of combining base learners' or estimators' predictions to improve model performance (Alelyani, 2021). Implementation Steps of Bagging:

- **Step 1:** Create multiple subsets from the original dataset with equal tuples, selecting observations with replacements.
- **Step 2:** Create a base model on each subset.
- **Step 3:** Each model is learned in parallel with each training set and independent of each other.
- **Step 4:** Averaging the predictions from all the models to calculate the final prediction.

To implement the ensemble, we use python libraries, including scikit-learn, etc. We have submitted all the codes as supplementary materials.

## 5 EXPERIMENTS AND EVALUATION

### 5.1 DATASETS

We have tested the proposed method on three different datasets. Here, we review the characteristics of the tested datasets:

- MNIST [37]: It's a handwritten digits database including a training set of 60,000 examples and a test set of 10,000 examples containing $28 \times 28$ grayscale images with ten classes.
- Fashion-MNIST [38]: It's a dataset of images consisting of a training set of 60,000 examples and a test set of 10,000 examples. Each example is a $28 \times 28$ grayscale image associated with a label from 10 classes.
- CIFAR10 [39] : This dataset consists of $32 \times 32$ color images in 10 classes, with 6000 images per class. There are 50000 training images and 10000 test images.

## 5.2 METRICS OF REPLACING ReLU WITH A POLYNOMIAL IN PRECISION, RECALL, AND F1-SCORE

In this section, we study the effect on Precision, Recall, and F1-score by replacing ReLU with a polynomial. Following the definition of :

**Precision:** It is the ratio of correctly predicted positive observations to the total predicted positive observation (Sokolova et al., 2006), (Cook & Ramadas, 2020), (Yacouby & Axman, 2020).

Precision = TP/(TP+FP)

**Recall:** The ratio of correctly predicted positive observations to all observations in the actual class observations to the total predicted positive observation (Sokolova et al., 2006), (Yacouby & Axman, 2020).

Recall = TP/TP+FN

**F1-score:** It is the weighted average of Precision and Recall. The score considers false positives and false negatives into account observations to the total predicted positive observation (Sokolova et al., 2006), (Yacouby & Axman, 2020).

F1 Score = 2*(Recall * Precision) / (Recall + Precision)

where True Positives (TP), True Negatives (TN), False Positives (FP) – False Negatives (FN) In this section, we calculate Precision, Recall, and F1-score for all three datasets using ReLU and polynomials for activation function; we study the effect of replacing ReLU in hidden layers in each class of datasets. Table 2 calculates Precision, Recall, and F1-score for MINST, Fashion-MNIST, and CIFAR10 by using ReLU and polynomial. Precision, Recall, and F1-score results drop while using polynomials in the hidden layers as an activation function, indicating a sharp performance decrease in the result. For instance, in class 1 in the dataset MINST, the F-1 score drops from 0.991 to 0.938 when using the polynomial.

Table 2: Calculate Precision, Recall, and F1-score for MINST, Fashion-MNIST, and CIFAR10 by using ReLU and polynomial where C is the class .

| | | | C-1 | C-2 | C-3 | C-4 | C-5 | C-6 | C-7 | C-8 | C-9 | C-10 |
|---|---|---|---|---|---|---|---|---|---|---|---|---|
| MNIST | ReLU | Precision | 0.986 | 0.986 | 0.981 | 0.994 | 0.982 | 0.974 | 0.911 | 0.990 | 0.986 | 0.994 |
| | | Recall | 0.996 | 0.998 | 0.972 | 0.959 | 0.987 | 0.927 | 0.930 | 0.991 | 0.968 | 0.992 |
| | | F1-Score | 0.991 | 0.992 | 0.948 | 0.976 | 0.984 | 0.952 | 0.951 | 0.980 | 0.958 | 0.968 |
| | Poly | Precision | 0.930 | 0.983 | 0.971 | 0.839 | 0.897 | 0.871 | 0.896 | 0.896 | 0.896 | 0.896 |
| | | Recall | 0.946 | 0.957 | 0.260 | 0.856 | 0.809 | 0.910 | 0.820 | 0.946 | 0.808 | 0.801 |
| | | F1-Score | 0.938 | 0.968 | 0.420 | 0.874 | 0.850 | 0.890 | 0.808 | 0.852 | 0.532 | 0.846 |
| F-MNIST | ReLU | Precision | 0.815 | 0.973 | 0.819 | 0.915 | 0.847 | 0.962 | 0.652 | 0.972 | 0.976 | 0.942 |
| | | Recall | 0.873 | 0.992 | 0.850 | 0.829 | 0.763 | 0.724 | 0.920 | 0.920 | 0.978 | 0.970 |
| | | F1-Score | 0.842 | 0.982 | 0.834 | 0.870 | 0.803 | 0.971 | 0.686 | 0.945 | 0.997 | 0.956 |
| | Poly | Precision | 0.804 | 0.969 | 0.920 | 0.871 | 0.162 | 0.961 | 0.108 | 0.875 | 0.813 | 0.662 |
| | | Recall | 0.638 | 0.921 | 0.441 | 0.669 | 0.839 | 0.773 | 0.590 | 0.693 | 0.888 | 0.983 |
| | | F1-Score | 0.739 | 0.944 | 0.944 | 0.773 | 0.272 | 0.857 | 0.138 | 0.774 | 0.849 | 0.762 |
| CIFAR_10 | ReLU | Precision | 0.724 | 0.825 | 0.548 | 0.553 | 0.643 | 0.673 | 0.750 | 0.761 | 0.851 | 0.871 |
| | | Recall | 0.790 | 0.879 | 0.713 | 0.507 | 0.614 | 0.569 | 0.805 | 0.781 | 0.799 | 0.847 |
| | | F1-Score | 0.756 | 0.850 | 0.620 | 0.530 | 0.627 | 0.617 | 0.777 | 0.771 | 0.824 | 0.813 |
| | Poly | Precision | 0.606 | 0.742 | 0.482 | 0.363 | 0.631 | 0.429 | 0.775 | 0.651 | 0.815 | 0.601 |
| | | Recall | 0.732 | 0.748 | 0.474 | 0.480 | 0.484 | 0.593 | 0.593 | 0.745 | 0.614 | 0.764 |
| | | F1-Score | 0.663 | 0.745 | 0.478 | 0.413 | 0.548 | 0.498 | 0.498 | 0.649 | 0.700 | 0.762 |

### 5.2.1 RESULT SEQUENTIAL ENSEMBLE TECHNIQUES

The Sequential ensemble techniques results are shown in Table 3, illustrating the resulting impact on accuracy. The result of sequential ensemble techniques can be summarized as follows:

- We measure the accuracy using polynomial and ReLU in the hidden layers. It decreases around 5%, 14%, and 10% in MINST, F-MINST, and CIFAR-10 when ReLU is replaced with a polynomial activation function.

- After applying the sequential ensemble approach, the accuracy increased about 4%, 8%, and 2% in MNIST, F-MNIST ,and CIFIAR-10.

- After five iterations, we successfully increased the accuracy and achieved performance at the same level as ReLU on average.

- The reason behind the higher accuracy is due to the implementation of the combination of models by aggregating the output from each model with the effects that reduce the model error and maintain the model's generalization.

- The sequential generation of base learners promotes the dependencies between the base learners. The model's performance is improved by assigning higher weights to previously misrepresented learners. Due to the sequential generation of the models, the time complexity is high. The overall delay will be the total time for all the sequential sub-model classifications.

Table 3 illustrates the resulting impact on accuracy using sequential ensemble techniques.

Table 3: The accuracy of the individual model with ReLU and polynomial activation function with sequential ensemble techniques where n = number of individual modes and n=1 polynomial without applying ensemble.

| | | MINST | F_MINST | CIFAR-10 |
|---|---|---|---|---|
| ReLU | | 99% | 89% | 83% |
| Polynomial | | 94% | 75% | 73% |
| Ensemble | N=2 | 95% | 83% | 75% |
| Number of individual model | N=4 | 96% | 86% | 77% |
| | N=6 | 99% | 89% | 83.2% |

subsectionResult of Parallel ensemble techniques The parallel ensemble's primary purpose is to reduce variance without increasing bias due to model averaging. The parallel ensemble techniques results can be summarized as follows:

- We study the impact of parallel bagging ensemble techniques on average expected loss, bias and variance.

- By replacing the ReLU with polynomial for both three datasets, the values in both bias and variance increase, which negatively affects the accuracy.

- We notice a significant improvement in all datasets even after three estimators in the variance reduction without increasing the bias, which is the expected result to decrease the error. For instance, in table 4 the variance value drops from 0.175 to 0.076 with a basis reduction from 1.748 to 1.130.

- We achieve the same average result as ReLU in variance and bias after 30 estimators without increasing bias. For instance,5 the variance value is 0.019, almost the same with or better than ReLU 0.038 for Minst and Fashion-MNIST datasets.

- We attain slight improvement after 100 estimates resulting in variance and bias. For instance, in table 5, the improvement is about 0.003, and no further improvement after that. There is a trade-off between time complexity and accuracy. As described earlier, our goal is to reduce bias and variance to increase accuracy performance.

- For CIFAR-10, it takes more time to reach the same result as ReLU about 100 estimators due to the complexity of the data table 6 illustrated the result.

- The main reason for improving the variance value in bagging is that dividing the data into different groups makes the data more distinguishable linearly. As a result, bagging can compensate for the performance gap between ReLU and polynomial activation functions in neural networks by decreasing the variance without increasing the bias.

- In parallel ensemble techniques, base learners are generated in a parallel format utilizing the parallel generation of base learners to encourage independence between the base learners. The independence of base learners significantly reduces the error due to the application of averages. In addition, due to the parallelism, execution will solve the sequential ensemble problem in time computing.

- The proposed bagging algorithm is experimentally proven to be very effective in improving classification accuracy. The results suggest that the bagging approach is very stable regarding feature selection due to the intrinsic power of reducing learning variance. Practically, the bagging concept creates new models, and the new mean shows a more actual picture of how those individual samples relate to each other in value. The more defined boundaries, the better the correlation between models, which reduces variance without increasing the bias. However, bagging slows down and grows more intensive as the number of iterations increases; clustered systems or a large number of processing cores will solve the problem of fast execution on large test sets.

Table 4, Table 5 and Table 6 illustrate that the bagging ensemble reduces the average expected bias and variance that accrued by using polynomial and bagging ensemble in MNIST, F-MNIST, and CIFAR-10.

Table 4: The average expected loss, bias, and variance for ReLU, polynomial, and bagging ensemble for Fashion-MNIST.

|  | | Average expected loss | Average bias | Average variance |
|---|---|---|---|---|
| ReLU | | 1.617 | 1.578 | 0.038 |
| Polynomial | | 1.700 | 1.784 | 0.175 |
| Ensemble Number of estimators | N=3 | 1.106 | 1.030 | 0.076 |
| | N=10 | 1.101 | 0.980 | 0.029 |
| | N=30 | 0.989 | 0.970 | 0.019 |
| | N=100 | 0.983 | 0.960 | 0.011 |

Table 5: The average expected loss, bias, and variance for ReLU, polynomial, and bagging ensemble for MNIST.

|  | | Average expected loss | Average bias | Average variance |
|---|---|---|---|---|
| ReLU | | 0.922 | 0.828 | 0.094 |
| Polynomial | | 1.300 | 1.124 | 0.175 |
| Ensemble Number of estimators | N=3 | 1.106 | 1.130 | 0.076 |
| | N=10 | 1.101 | 0.980 | 0.029 |
| | N=30 | 0.989 | 0.970 | 0.019 |
| | N=100 | 0.981 | 0.965 | 0.016 |

Table 6: The average expected loss, bias, and variance for ReLU, polynomial, and bagging ensemble for CIFAR-10.

|  | | Average expected loss | Average bias | Average variance |
|---|---|---|---|---|
| ReLU | | 7.483 | 7.381 | 0.102 |
| Polynomial | | 11.441 | 9.816 | 1.625 |
| Ensemble Number of estimators | N=3 | 8.687 | 8.643 | 0.043 |
| | N=10 | 8.232 | 8.211 | 0.041 |
| | N=100 | 7.642 | 7.575 | 0.032 |

Our goal is not to outperform the state-of-the-art non-HE results on MNIST, Fashion-MNIST, and CIFAR-10. We aim to solve the accuracy problem and prove that the ensemble can close the accuracy gap caused by replacing the ReLU function with polynomial activation and improve the accuracy significantly to achieve equal performance as ReLU. Furthermore, the ensemble can be combined with other approaches, such as batch normalization, which makes the network more robust against gradient vanishing or exploding (Bjorck et al., 2018).

## 6 CONCLUSION

In this work, we apply an ensemble approach to solve the problem of decreasing accuracy when replacing the ReLU with a polynomial activation function in the hidden layers in HE-based ML. HE scheme cannot support comparison, division, and exponential operations in a straightforward manner; some functions used in deep learning, such as ReLU, Sigmod, and max-pooling, are incompatible with HE. Polynomial activation functions are considered to address this issue because an HE scheme can straightforwardly support polynomial-based calculations. However, using polynomials as an activation function causes low accuracy. We propose to apply the ensemble approach to solve the accuracy problem caused by replacing ReLU with a low-degree polynomial. Our experiment results show increased accuracy in three datasets and achieve the same performance as ReLU in parallel and sequential ensemble techniques.

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
