# OpenReview forum: "Ensemble Homomorphic Encrypted Data Classification"
_ICLR.cc/2023/Conference — Submitted to ICLR 2023_

### Official Review · Reviewer_kymY · 2022-10-16

**Confidence:** 2
**Correctness:** 3
**Technical Novelty And Significance:** 2
**Empirical Novelty And Significance:** 1
**Recommendation:** 3

**Clarity, Quality, Novelty And Reproducibility:**

Clarity: The manuscript describe issue and solution quite straightforwardly, so yes, it is very clear for readers.

Quality: The writing is too simple and only combines existing concepts together.

Novelty: Neither replacing relu with polynomial nor ensemble in HE is innovative and this work merely combines them.

**Strength And Weaknesses:**

Strength:

1. Separating ensemble into sequential and parallel manner is good to take advantages of both approaches.

2. This work shows limitation of ensemble instead of overrating it.

Weakness:

1. This work uses relatively shallow NN compared to existing HE-based DL methods.

2. Lack of discussion on threat/security model since this work is based on cryptographic setting.

3. Datasets used in this work are quite out-dated in the ML community.

**Summary Of The Paper:**

Homomorphic encryption allows to compute over encrypted data outsourcing to cloud servers securely. Hence, HE can be applied to deep learning by scaling efficiently and cost less. Since activation like ReLU is not compatible with HE, HE can apply polynomial activation function with merely addition and multiplication. Also, the lower-degree polynomial suffers from high bias, variance and low accuracy, so this work applies ensemble methods to enhance outputs from multiple weak learners.

**Summary Of The Review:**

This work is less innovative. It is merely combines ensemble and polynomial replacement of relu together, and tests on relatively simple datasets. To be honest, I am not sure if there is a significant impact in this work on HE-based ML.

---

### Official Review · Reviewer_Kc3a · 2022-10-19

**Confidence:** 4
**Correctness:** 3
**Technical Novelty And Significance:** 1
**Empirical Novelty And Significance:** 1
**Recommendation:** 1

**Clarity, Quality, Novelty And Reproducibility:**

The paper is not written to a high standard with frequent grammatical errors, making it hard to follow. The paper studies an interesting and topic, but unfortunately the main results are to be expected by anyone familiar with the field. There is code attached to reproduce the results, although I have not tried to run it.

**Strength And Weaknesses:**

Strengths:
- Evaluating neural networks under FHE is a timely topic worthy of research.
- The main hypothesis studied in this paper (that ensembling techniques improve accuracy) seems to be validated by the experimental results.

Weaknesses:
The properties of ensembling techniques that are investigated here are widely known and can be found in any machine learning textbook. The main result here is that ensembling methods can improve accuracy by reducing bias and/or variance, which is by itself not worthy of publication even if it is applied in the FHE domain. I encourage the authors to view these results are a good starting point for further investigation. Some interesting research directions from here could be:
1. How do ensembling techniques compare with other techniques for improving the accuracy in FHE domain such as increasing the degree of the polynomial approximation to ReLU?
2. How do these methods compare in terms of training time and inference latency?
3. What is the effect of using the same ensembling techniques in the non-encrypted domain? Do we see the same accuracy improvement or do these techniques somehow work especially well due to the approximation error in FHE?
4. Were these neural networks actually implemented using an FHE library? From a quick glance at the code provided, it looks not. How would the accuracy and runtime be affected? Remember that FHE schemes for floating point arithmetic like CKKS are inherently noisy and may lead to different results (beyond just the effect of replacing the ReLU with a polynomial).



**Summary Of The Paper:**

This paper studies the effect of ensembling techniques (sequential and parallel) on neural networks in the fully homomorphic encrypted (FHE) domain. In order to evaluate the neural networks using FHE, the ReLU activation function is approximated by a degree-2 polynomial. The authors find that this leads to a significant drop in accuracy. The authors verify that applying either sequential or parallel ensembling techniques can improve the accuracy.

**Summary Of The Review:**

The finding that ensembling techniques improve accuracy of neural networks, that use a degree 2 polynomial approximation to ReLU, is not by itself worthy of publication. However, this finding could an interesting starting point for further research in the area.

---

### Official Review · Reviewer_WXCk · 2022-10-24

**Confidence:** 5
**Correctness:** 2
**Technical Novelty And Significance:** 1
**Empirical Novelty And Significance:** 1
**Recommendation:** 1

**Clarity, Quality, Novelty And Reproducibility:**

* The level of novelty is pretty negligible. More comprehensive analysis of ensemble approaches in HE domain is needed.
* The quality of the writing is low. There are many grammatical errors. The manuscripts needs a significant refinement in the exposition.

**Strength And Weaknesses:**

Weaknesses:
1 - The main idea of using ensemble of neural networks is trivial and very common in machine learning literature. The paper doesn't provide any specific adaptation to the homomorphic encryption domain.
2 - The discussion on the homomorphic encryption schemes is completely missing. What type of HE do you use?
3 - How do you preform majority voting in the encrypted domain? Most of HE schemes do not support argmax operation.
4 - For sequential ensembling, it is important to study the effect of noise accumulation in the context of homomorphic encryption. This limitations prevents the use of even single deep neural networks on homomorphically encrypted data.

**Summary Of The Paper:**

This paper studies the impact of ensemble modeling on the performance of homomorphic encryption compatible neural networks.

**Summary Of The Review:**

Overall, I recommend against the acceptance of this paper due to:
1 - low novelty (trivial ensembling)
2 - analysis on the most important aspects of HE is missing

---

### Official Review · Reviewer_pCgq · 2022-10-24

**Confidence:** 4
**Clarity, Quality, Novelty And Reproducibility:** The lack of scientific novelty in the…
**Correctness:** 4
**Technical Novelty And Significance:** 1
**Empirical Novelty And Significance:** 1
**Recommendation:** 1

**Strength And Weaknesses:**

This is a strange paper. It appears to propose to use the well-known ensemble learning techniques with neural networks as base models. The first question that pops to my mind is – well, isn’t an ensemble of neural networks simply a wider (or deeper) neural network?

Ignoring this – the paper is full of known machine learning (see for example Section 3.3 where they describe the advantages/disadvantages of ensemble learning) but there is really nothing new. Certainly not worthy of a top-conference such as ICLR.


**Summary Of The Paper:**

The paper proposes to study the impact of ensemble learning on neural networks with polynomial approximations of the ReLU function. The motivation of using polynomial approximations comes from the need to apply Homomorphic encryption on inference (this is a standard technique in privacy preserving ML under homomorphic encryption).

**Summary Of The Review:**

The paper applies ensemble techniques with neural networks as base models, with a caveat that the activation functions of the neural networks are polynomials approximating ReLU. I don't see anything novel here -- instead it is a rather weak engineering experiment.

---

### Decision · Program_Chairs · 2023-01-20

**Decision:**

Reject

**Justification For Why Not Higher Score:**

No response.

**Justification For Why Not Lower Score:**

NA

**Metareview: Summary, Strengths And Weaknesses:**

The reviews for the paper are negative and the authors have not participated in the discussion.